# Hypoxia ameliorates intestinal inflammation through NLRP3/mTOR downregulation and autophagy activation

Jesus Cosin-Roger[1], Simona Simmen[1], Hassan Melhem[1], Kirstin Atrott[1], Isabelle Frey-Wagner[1], Martin Hausmann[1], Cheryl de Vallière[1], Marianne R. Spalinger[1], Patrick Spielmann[2,3], Roland H. Wenger[2,3], Jonas Zeitz[1], Stephan R. Vavricka[1], Gerhard Rogler[1,3] & Pedro A. Ruiz[1]

Hypoxia regulates autophagy and nucleotide-binding oligomerization domain receptor, pyrin domain containing (NLRP)3, two innate immune mechanisms linked by mutual regulation and associated to IBD. Here we show that hypoxia ameliorates inflammation during the development of colitis by modulating autophagy and mammalian target of rapamycin (mTOR)/NLRP3 pathway. Hypoxia significantly reduces tumor necrosis factor α, interleukin (IL)-6 and NLRP3 expression, and increases the turnover of the autophagy protein p62 in colon biopsies of Crohn's disease patients, and in samples from dextran sulfate sodium-treated mice and Il-10−/− mice. In vitro, NF-κB signaling and NLRP3 expression are reduced through hypoxia-induced autophagy. We also identify NLRP3 as a novel binding partner of mTOR. Dimethyloxalylglycine-mediated hydroxylase inhibition ameliorates colitis in mice, downregulates NLRP3 and promotes autophagy. We suggest that hypoxia counteracts inflammation through the downregulation of the binding of mTOR and NLRP3 and activation of autophagy.

[1] Department of Gastroenterology and Hepatology, University Hospital Zurich, University of Zurich, Rämistrasse 100, 8091 Zurich, Switzerland. [2] Institute of Physiology, University of Zurich, Winterthurerstrasse 190, 8057 Zurich, Switzerland. [3] Zurich Center for Integrative Human Physiology (ZIHP), University of Zurich, 8057 Zurich, Switzerland. Correspondence and requests for materials should be addressed to P.A.R. (email: PedroAntonio.Ruiz-Castro@usz.ch)

Crohn's disease (CD) and ulcerative colitis (UC), the main clinical phenotypes of inflammatory bowel diseases (IBD), are complex diseases arising from the interplay between genetic and environmental factors that influence mucosal homeostasis and trigger an inappropriate immune response[1, 2]. In recent years, environmental hypoxia has been increasingly recognized as an important environmental factor influencing the development of IBD[3, 4]. Because of the integral role of the epithelium in maintaining intestinal homeostasis, intestinal epithelial cells (IECs) have evolved many molecular mechanisms to cope with fluctuations in blood flow, oxygen availability and metabolism, resulting in tissue hypoxia, particularly during the severe shifts associated with active inflammation in IBD[5, 6]. Adaptive transcriptional responses to oxygen deprivation are

**Fig. 1** Hypoxia reduces inflammatory gene expression and induces autophagy flux in *CD* patients. **a**, **b**, **c**, **d**, **e**, and **f** Healthy volunteers (*HV*, n = 10), patients with Crohn's disease (*CD*, n = 11) and patients with ulcerative colitis (*UC*, n = 9) were subjected to hypoxic conditions in a hypobaric chamber simulating an altitude of 4000 m.a.s.l. for 3 h. Distal colon biopsies were taken the day before entering the hypobaric chamber (T0), immediately after hypoxia (T1), and 1 week after collection of the first biopsy (T2), and transcript analysis was performed. Statistical analysis was performed using one-way ANOVA followed by Tukey's post-test. Results represent mean + s.e.m.,*P < 0.05; **P < 0.01; ***P < 0.001; *ns* = not significant. **g** Immunostaining for p62 (*arrowheads*) was performed on distal colon sections and immunohistochemistry (IHC) score was determined by two independent, blinded investigators. Scale bar, 25 μm

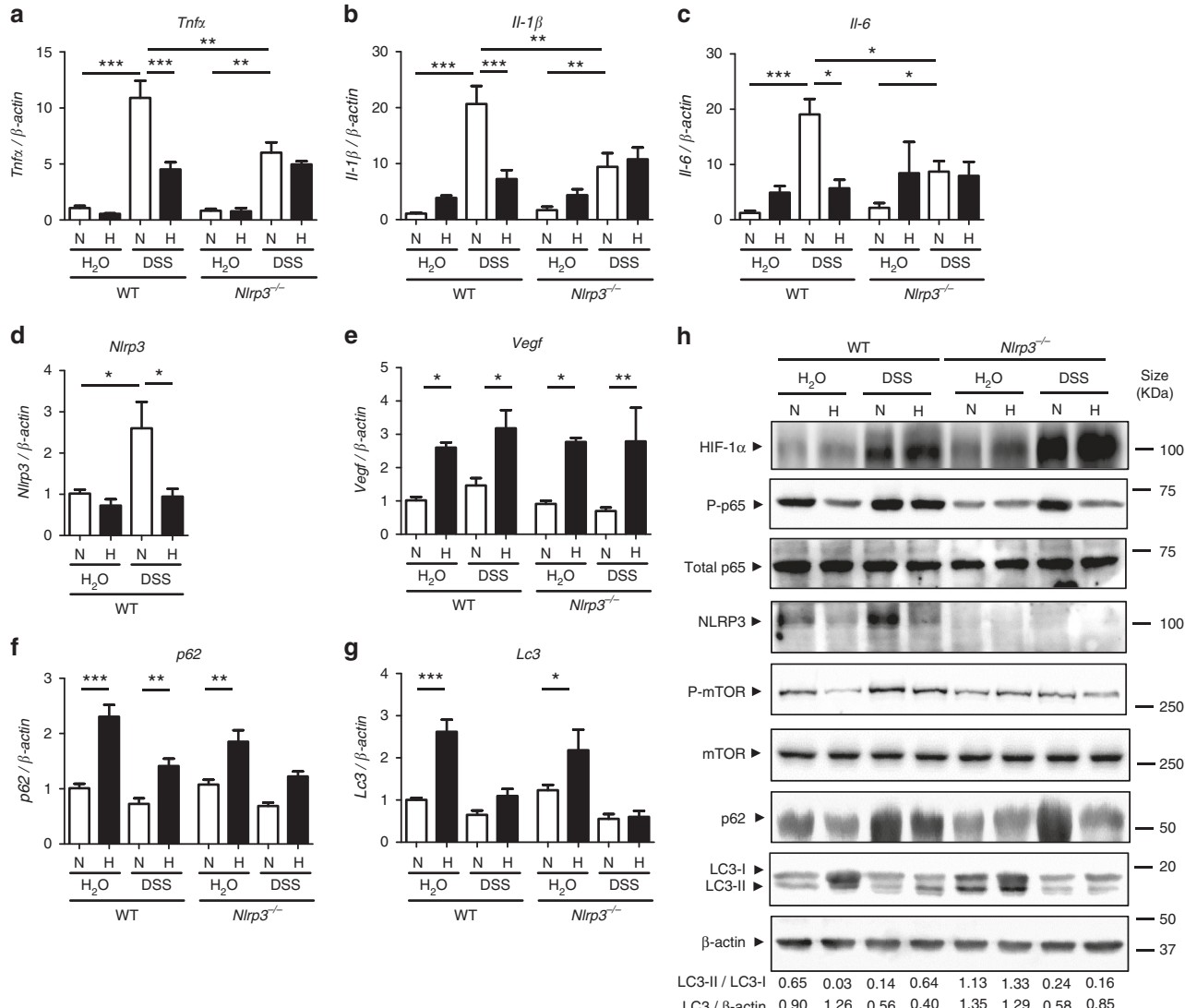

**Fig. 2** Hypoxia reduces inflammatory processes and induces autophagy in the DSS mouse model of colitis. **a**, **b**, **c**, **d**, **e**, **f** and **g** WT and $Nlrp3^{-/-}$ mice were administered with DSS or DSS-free water and subjected to normoxia (N, 21% $O_2$) or hypoxia (H, 8% $O_2$). After 18 h, mice were killed and colon biopsies were collected. Statistical analysis was performed using one-way ANOVA followed by Tukey's post-test. Results represent mean + s.e.m., WT mice under normoxia: $n = 5$, WT mice under normoxia: $n = 5$, DSS-treated WT mice under normoxia: $n = 6$; $Nlrp3^{-/-}$ mice under normoxia: $n = 5$; DSS-treated $Nlrp3^{-/-}$ mice under normoxia: $n = 6$; WT mice under hypoxia: $n = 4$, DSS-treated WT mice under hypoxia: $n = 5$; $Nlrp3^{-/-}$ mice under hypoxia: $n = 3$; DSS-treated $Nlrp3^{-/-}$ mice under hypoxia: $n = 3$, *$P < 0.05$; **$P < 0.01$; ***$P < 0.001$. **h** Total protein was isolated and western blot performed. LC3-I and LC3-II bands were quantified, and autophagy was measured by variations in the ratio of LC3-II/LC3-I and the total amount of LC3 (LC3-I plus LC3-II) relative to β-actin

mediated mainly through the hypoxia inducible factor (HIF) complexes, which are tightly regulated through the action of three prolylhydroxylases and one asparagine hydroxylase[7]. In steady state conditions, hydroxylases suppress the activity of HIF through the hydroxylation of Pro402 and Pro564 of HIF-1α, leading to ubiquitination and proteasomal degradation[8]. A further hydroxylation of Asn803 prevents HIF transcriptional activity. Hypoxia, blocks these processes, allowing HIF-1α to accumulate and induce the expression of HIF target genes[9]. Both HIF-1α and HIF-2α are induced in the inflamed mucosa from IBD patients and mouse models of colitis[10, 11], and promote epithelial barrier function and the resolution of inflammation in various mouse models of disease[5, 12, 13]. Hypoxia activates further innate immune protective mechanisms such as autophagy. Thus, HIF-1α induces the expression of several autophagosome markers, including LC3, autophagy related gene (ATG)5 and beclin-1, as well as the conversion of LC3-I to LC3-II[14].

Functional polymorphisms in gene loci, including autophagy proteins such as ATG16L1, IRGM, and nucleotide-binding oligomerization domain-containing protein (NOD)2 are associated with an increased risk of CD, suggesting defective autophagy in the pathogenesis of IBD[15, 16]. Autophagy ensures energetic homeostasis in cells by providing nutrients during metabolic stress and also serves a protective role by removing damaged components from the cytosol[17]. Autophagic adaptor proteins, such as p62 (also known as SQSTM1/sequestosome 1), deliver target substrates for degradation to the autophagosome via binding to the microtubule-associated protein 1 light chain 3 (LC3)[18]. LC3 is linked to phosphatidylethanolamine, yielding a product known as LC3-II[19], which remains associated with autophagosomes until its destruction. Thus, the amount of membrane-bound LC3-II provides a good index of autophagy induction[20]. Environmental stress induces autophagy through the inhibition of mammalian target of rapamycin (mTOR), a serine/

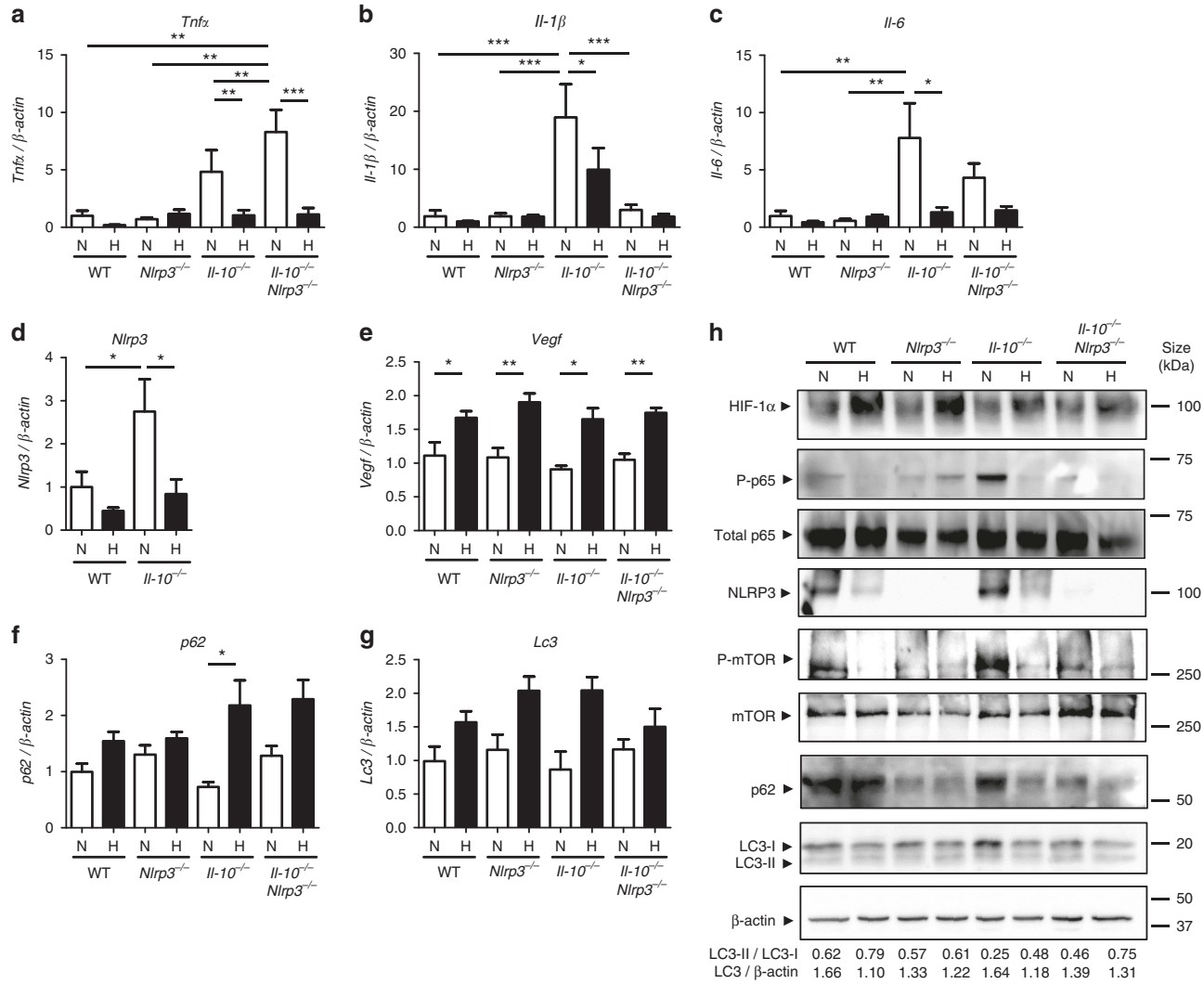

**Fig. 3** Hypoxia reduces inflammatory processes and induces autophagy in *Il-10^{-/-}* mice. **a**, **b**, **c**, **d**, **e**, **f** and **g** WT, *Nrlp3^{-/-}*, *Il-10^{-/-}*, and *Il-10^{-/-}Nrlp3^{-/-}*double knockout mice were subjected to normoxia (*N*, 21% O$_2$) or hypoxia (*H*, 8% O$_2$). After 18 h, mice were killed and colon biopsies were collected. Statistical analysis was performed using one-way ANOVA followed by Tukey's post-test. Results represent mean + s.e.m., WT mice under normoxia: *n* = 5, WT mice under hypoxia: *n* = 5; *Nrlp3^{-/-}* mice under normoxia: *n* = 4; *Nrlp3^{-/-}* mice under hypoxia: *n* = 5; *Il-10^{-/-}* mice under hypoxia: *n* = 4; *Il-10^{-/-}* mice under hypoxia: *n* = 4; *Il-10^{-/-}Nrlp3^{-/-}* mice under normoxia: *n* = 7; *Il-10^{-/-}Nrlp3^{-/-}* mice under hypoxia: *n* = 9, *$P < 0.05$; **$P < 0.01$; ***$P < 0.001$. **h** Total protein was isolated and western blot performed. LC3-I and LC3-II bands were quantified, and autophagy was measured by variations in the ratio of LC3-II/LC3-I and the total amount of LC3 (LC3-I plus LC3-II) relative to β-actin

threonine protein kinase that belongs to the phosphatidylinositol 3-kinase-related kinase family[21, 22]. Hypoxia inhibits mTOR signaling through HIF-1-dependent and -independent mechanisms, as well as through proteins that prevent the binding of mTOR with the mTOR upstream activator Ras homolog enriched in brain (RHEB)[23]. Despite recent efforts, the molecular mechanisms underlying the induction of autophagy in response to hypoxia remain incompletely understood.

Hypoxia has also been associated with the activation of NOD-like receptor, pyrin domain-containing (NLRP)3. NLRP3 is best known as a component of the inflammasome, a multiprotein signaling complex responsible for the activation of caspase-1 and subsequent maturation of pro-interleukin (IL)-1β and pro-IL-18 that can be activated by a wide range of stimuli, like adenosine triphosphate (ATP)[24], reactive oxygen species (ROS)[25], and inorganic materials[26, 27]. NLRP3 and the inflammasome pathways play a decisive role in maintaining intestinal homeostasis, and single-nucleotide polymorphisms in the *NLRP3* gene have recently been associated with the development of CD[28].

Accumulating evidence indicates that NLRP3 and autophagy pathways are linked by reciprocal regulation, and recent studies have shown autophagy-dependent degradation of NLRP3[29, 30].

Our results identify NLRP3 as a binding partner of mTOR, and show that environmental hypoxia ameliorates intestinal inflammation through the reduction of NLRP3 and mTOR binding, and the subsequent activation of autophagy.

## Results

**Hypoxia reduces inflammatory gene expression in CD patients.** To study the effects of hypoxia on colonic inflammatory pathways, healthy subjects and patients with IBD in clinical remission were subjected to hypoxic conditions, resembling an altitude of 4000 m above sea level (m.a.s.l.) for 3 h. Transcriptional analysis of distal colon biopsies from patients with CD, but not UC, evidenced increased levels of NLRP3 and tumor necrosis factor (TNF)α, but not IL-1β or IL-6 mRNA prior to hypoxia (Fig. 1a–d). Hypoxia significantly reduced the mRNA expression of TNFα (Fig. 1a), IL-6 (Fig. 1c) and NLRP3 (Fig. 1d)

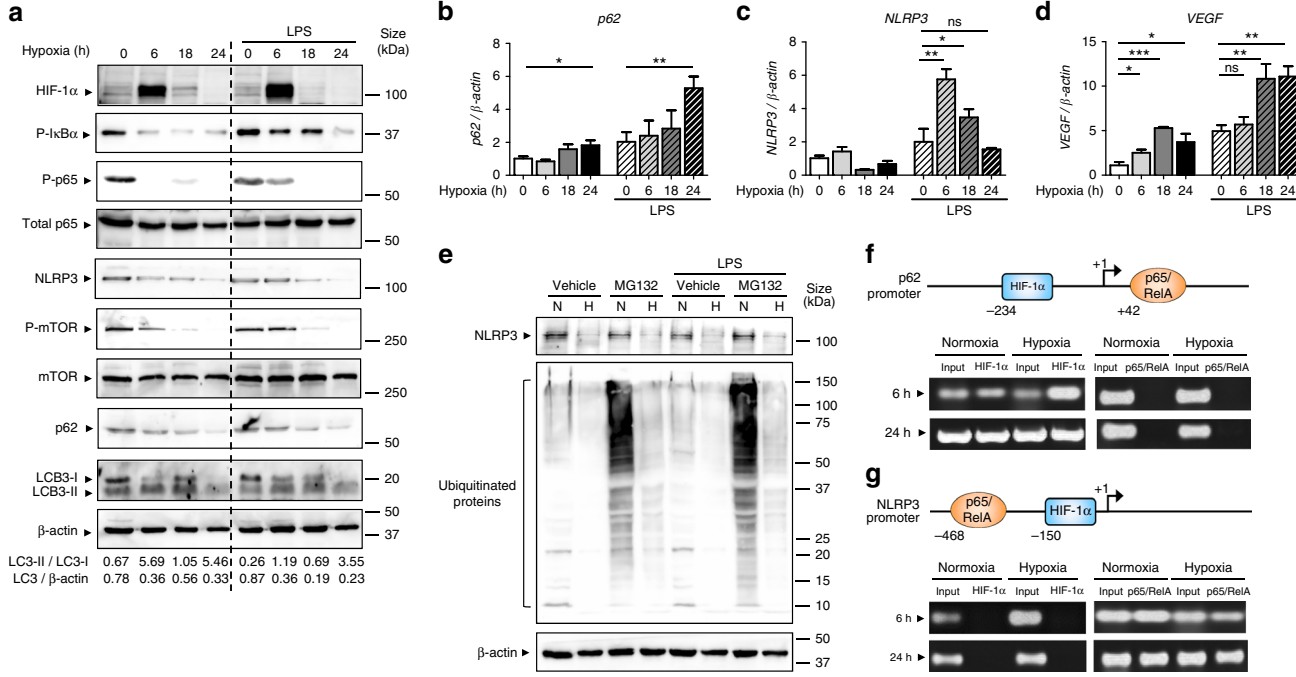

**Fig. 4** Hypoxia reduces inflammatory signaling pathways and NLRP3 expression and induces autophagy in IECs. **a** HT-29 cells were subjected to normoxia and hypoxia at the indicated times in the absence or presence of 10 µg/ml LPS. Autophagy was measured by variations in the ratio of LC3-II/LC3-I and the total amount of LC3 (LC3-I plus LC3-II) relative to β-actin. Results are representative of two independent experiments. **b**, **c** and **d** HT-29 cells were subjected to normoxia and hypoxia for the indicated periods in the absence or presence of 10 µg/ml LPS, followed by transcript analysis. Statistical analysis was performed using one-way ANOVA followed by Tukey's post-test. Results represent mean + s.e.m. of two independent experiments done in triplicate, *$P < 0.05$; **$P < 0.01$; ***$P < 0.001$; *ns* not significant. **e** HT-29 cells were subjected to normoxia or hypoxia in the presence and absence of 10 µg/ml LPS and 20 µM of MG132. Results are representative of two independent experiments. **f** and **g** Putative binding sites for HIF-1α and NF-κB were found in the p62 **f** and NLRP3 **g** promoters using Genomatix software tools. Numbers under the boxes indicate the distance from the transcription start site. HT-29 cells were subjected to normoxia (21% $O_2$) or hypoxia (0.2% $O_2$) for 6 h and 24 h. ChIP analysis was performed using antibodies against HIF-1α and NF-κB for immunoprecipitation. PCR was performed using the promoter-specific primers for the p62 **f** and NLRP3 **g** promoter binding sites of HIF-1α and NF-κB. Aliquots taken prior to immunoprecipitation were used as input control. PCR products were run on 2% agarose gel. The results are representative of three independent experiments.

immediately after hypoxia, and this reduction was still evident 1 week later. Interestingly, hypoxia downregulated inflammatory gene expression concomitantly with the induction of autophagy-associated p62 (Fig. 1e), but not LC3 (Fig. 1f). Although histology samples did not show any differences in the inflammatory status of patients with IBD prior to or after hypoxia, immunostaining for p62 of distal colon sections showed a trend towards reduced protein levels of p62 1 week after hypoxia in HV and patients with CD, indicating a hypoxia-mediated increase in autophagy flux (Fig. 1g, *arrowheads*).

**Hypoxia reduces DSS-induced inflammatory gene expression.** To further study the influence of hypoxia in inflammation, and the role of NLRP3 in hypoxia-mediated effects, WT and $Nlrp3^{-/-}$ mice were administered with 2% dextran sulfate sodium (DSS) for 7 days in drinking water and subjected to normoxia or hypoxia for 18 h. DSS-treated WT and $Nlrp3^{-/-}$ mice presented increased mRNA expression of proinflammatory mediators, including $Tnf\alpha$, $Il$-6, $Il$-1$\beta$, and $Nlrp3$ (Fig. 2a–d). Hypoxia significantly reduced $Tnf\alpha$, $Il$-6, $Il$-1$\beta$, and $Nlrp3$ mRNA levels in WT mice, but not $Nlrp3^{-/-}$ mice (Fig. 2a–d), supporting a role for NLRP3 in hypoxia-mediated inhibitory effects. Hypoxia also induced the expression of the autophagy proteins p62 and $Lc3$ in both WT and $Nlrp3^{-/-}$ mice (Fig. 2f and g). Hypoxic conditions were confirmed through the induction of the hypoxia marker vascular endothelial growth factor ($Vegf$) (Fig. 2e). Under normoxia, DSS-treated mice presented increased levels of

phosphorylated NF-κB subunit p65/RelA concomitantly with p62 accumulation (Fig. 2h and Supplementary Fig. 1A). In WT mice, hypoxia reversed DSS-induced p65/RelA phosphorylation and NLRP3 expression, and restored autophagy as evidenced by the reduction of mTOR phosphorylation and degradation of p62, as well as the increase of the ratio LC3-II/LC3-I and LC3 degradation (Fig. 2h and Supplementary Fig. 1B–D). Interestingly, $Nlrp3^{-/-}$ mice showed reduced levels of phosphorylated mTOR when compared to WT mice as well as impaired responses to hypoxia evidenced by the lack of marked changes in the ratio LC3-II/LC3-I and LC3 degradation (Fig. 2h and Supplementary Fig. 1C, D), suggesting a role for NLRP3 in the control of autophagy.

**Hypoxia blocks inflammatory gene expression in $Il$-$10^{-/-}$ mice.** Next, we sought to confirm the anti-inflammatory effects of hypoxia in the $Il$-$10^{-/-}$ mouse model of colitis. WT, $Il$-$10^{-/-}$, $Nlrp3^{-/-}$, and $Il$-$10^{-/-}Nlrp3^{-/-}$ double knockout mice were subjected to normoxia or hypoxia for 18 h prior to colon biopsy collection. Under normoxic conditions, $Il$-$10^{-/-}$ mice presented increased mRNA levels of $Tnf\alpha$, $Il$-6, $Il$-1$\beta$ and $Nlrp3$, which were significantly reduced under hypoxia (Figs. 3a–d), together with the induction of p62 expression (Fig. 3f). Although not-significant, our results also show a clear trend towards an increase in the mRNA levels of $Lc3$ under hypoxia (Fig. 3g). Under normoxia, $Il$-$10^{-/-}$ mice also presented an increased accumulation of phosphorylated p65/RelA, phosphorylated mTOR, p62 and

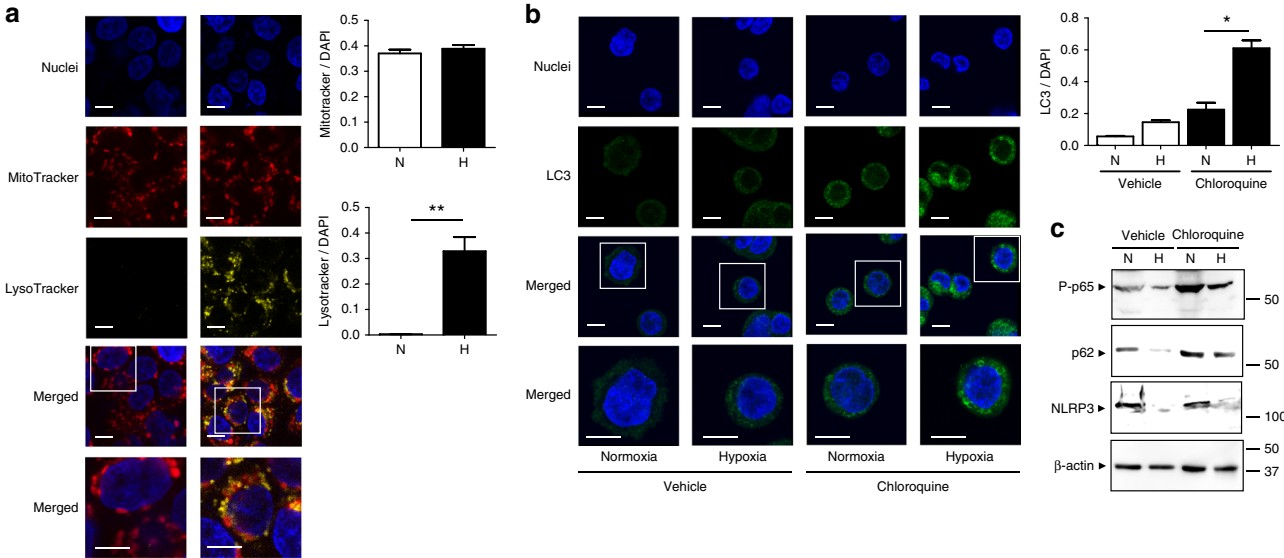

**Fig. 5** Hypoxia reduces NF-κB activation through the activation of autophagy. **a** HT-29 cells were subjected to hypoxia (0.2% O$_2$, 24 h) and stained with DAPI (*blue*), as well as LysoTracker Yellow-HCK-123 (*yellow*) or MitoTracker Deep Red FM (*red*) to monitor lysosomal accumulation and functional mitochondria, respectively. Examples of cells containing both markers are presented in the magnified insets. Scale bar, 10 μm. Changes in lysosomal accumulation and functional mitochondria were calculated relative to DAPI staining from at least 10 areas of interest pooled from three independent experiments by two independent, blinded investigators. Statistical analysis was performed using Student *t*-test. Results represent mean + s.e.m., **$P < 0.01$. **b** HT-29 cells were subjected to normoxia (21% O$_2$) or hypoxia (0.2% O$_2$) for 24 h in the presence or absence of 40 μM chloroquine. Cells were stained for LC3 (*green*) or cell nuclei (DAPI (*blue*)). Examples of cells containing both markers are presented in the magnified insets. Scale bars, 10 μm. Changes in LC3 accumulation were calculated relative to DAPI staining from at least 10 areas of interest pooled from four independet experiments. Statistical analysis was performed using one-way ANOVA followed by Tukey's post-test. Results represent mean + s.e.m., *$P < 0.05$. **c** LPS-treated HT-29 cells were subjected to normoxia (21% O$_2$) or hypoxia (0.2% O$_2$) for 24 h in the presence or absence of 40 μM chloroquine. Results are representative of two independent experiments

LC3 compared to *Il-10*$^{-/-}$ *NLRP3*$^{-/-}$ mice (Fig. 3h and Supplementary Fig. 2A, C–E), suggesting a blockage of autophagy promoted by NLRP3 concomitantly with the induction of the NF-κB signaling pathway. Conversely, and in line with our results in the DSS mouse model, hypoxia reduced NLRP3 expression and p65/RelA phosphorylation, and restored autophagy as evidenced by the reduction of phosphorylated mTOR, the increase of the ratio LC3-II/LC3-I and subsequent degradation of LC3 and p62 (Fig. 3h and Supplementary Fig. 2A–E).

**Hypoxia induces NLRP3 degradation and autophagy in IECs.** Since our results in patients with CD and the DSS and *Il-10*$^{-/-}$ mouse models of colitis suggest a protective effect of hypoxia in the colonic mucosa linked to the reduction of NLRP3 and the activation of autophagy, we sought to elucidate the molecular mechanisms governing the regulation of these pathways in IECs. For this purpose, we subjected the human IEC line HT-29 to hypoxia in the presence and absence of lipopolysaccharide (LPS). HIF-1α protein accumulation peaked after 6 h of hypoxia (Fig. 4a and Supplementary Fig. 3A). In line with our results in mice, constitutive phosphorylation of NF-κB subunit p65/RelA and the NF-κB upstream inhibitor IκBα was reversed under hypoxia in a time-dependent manner (Fig. 4a and Supplementary Fig. 3B, C). Hypoxia also activated autophagy, as evidenced by the reduction of the phosphorylation/activation of mTOR, the conversion of LC3-I into LC3-II, and the subsequent degradation of p62 and LC3 protein (Fig. 4a and Supplementary Fig. 3E–G). Interestingly, hypoxia also reduced constitutive and LPS-induced protein expression of NLRP3 (Fig. 4a and Supplementary Fig. 3D). In accordance with our *in vivo* results, hypoxia triggered p62 mRNA expression (Fig. 4b). Likewise, hypoxia induced a transient

expression of NLRP3 mRNA after 6 and 18 h, which returned to basal levels after 24 h of hypoxia (Fig. 4c), indicating that NLRP3 protein expression was not due to a reduction in the levels of NLRP3 mRNA. Protein levels of NLRP3 were also reduced in the presence of the proteasome inhibitor MG132, revealing that hypoxia induces NLRP3 clearance through a proteasome-independent mechanism (Fig. 4e and Supplementary Fig. 3H). Interestingly, hypoxia reversed the accumulation of ubiquitinated proteins in the presence of the proteasome inhibitor, pointing to the activation of alternative protein degradation mechanisms under hypoxia such as autophagy (Fig. 4e).

Following our results on hypoxia-mediated regulation of p62 and NLRP3 expression, we sought to elucidate the molecular mechanisms governing the transcriptional regulation of both genes. Sequence analysis of the p62 promoter identified putative binding sites for HIF-1α and p65/RelA at −234 and +42, respectively (Fig. 4f). One putative binding site for p65/RelA was also found at −468 and one binding site for HIF-1α at −150 in the NLRP3 promoter (Fig. 4g). To investigate the binding activity of both transcription factors to the p62 and NLRP3 promoters in HT-29 cells subjected to hypoxia (0.2 % O$_2$, 6 and 24 h), chromatin immunoprecipitation (ChIP) analysis was performed. In line with the stabilization of HIF-1α after 6 h of hypoxia, ChIP analysis shows increased biding of HIF-1α to the promoter of p62 after 6 h of hypoxia compared to normoxia, suggesting a role for HIF-1α in the induction of p62 expression observed under low oxygen level conditions (Fig. 4f). In accordance with the reduction in p65/RelA phosphorylation under hypoxia, binding of NF-κB to the p62 promoter was not induced at low oxygen levels (Fig. 4g). In contrast, and in line with the constitutive expression of NLRP3 in unstimulated cells, we observed constitutive binding of NF-κB to the NLRP3

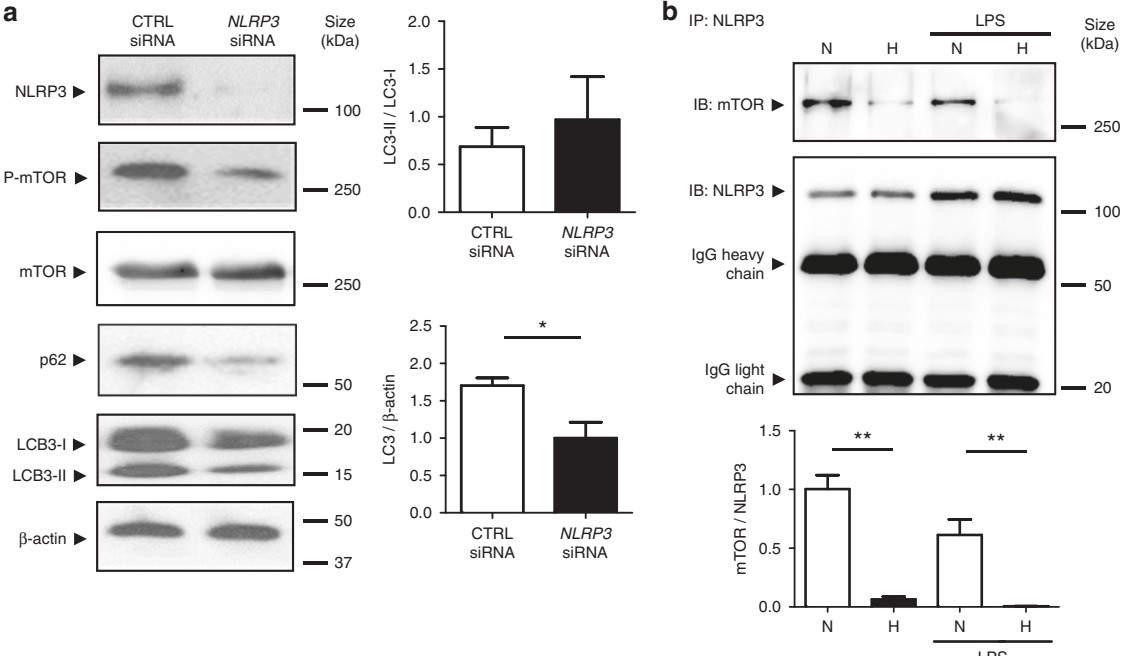

**Fig. 6** NLRP3 regulates autophagy through an inflammasome-independent mechanism via direct binding to mTOR. **a** HT-29 cells were transfected with NLRP3-specific siRNA or negative control siRNA using Lipofectamine RNAiMAX. Forty-eight hours after transfection, total protein was isolated and western blot performed. Quantification of the ratio of LC3-II/LC3-I and the total amount of LC3 (LC3-I plus LC3-II) relative to β-actin is presented. Statistical analysis was performed using Student *t*-test. Results represent mean + s.e.m. of three independent experiments, *$P < 0.05$. **b** HT-29 cells were subjected to normoxia (21% $O_2$) or hypoxia (0.2% $O_2$) for 24 h in the presence and absence of LPS. Co-IP using NLRP3 antibody was performed, followed by anti-mTOR immunoblotting. Quantification relative to cells subjected to normoxia in the absence of LPS after normalization to NLRP3 is presented. Statistical analysis was performed using one-way ANOVA followed by Tukey's post-test. Results represent mean of three independent experiments + s.e.m., **$P < 0.01$

promoter. Of note, HIF-1α was not found in the promoter of NLRP3 despite the accumulation of HIF-1α observed during hypoxia (Fig. 4g).

**Hypoxia-mediated autophagy reduces NF-κB signaling in IECs.** Biosynthesis and accumulation of lysosomes in HT-29 cells under hypoxia was monitored using LysoTracker Yellow-HCK-123, a fluorescence probe that accumulates in acidic organelles such as lysosomes. Hypoxia induced the accumulation of lysosomes, further confirming the induction of autophagy at low oxygen levels. Active mitochondria monitored using MitoTracker Deep Red FM were not affected after 24 h of hypoxia (Fig. 5a). In order to clarify the causality between hypoxia-mediated autophagy and the reduction of inflammatory processes, HT-29 cells were subjected to hypoxia in the presence of chloroquine, a late-stage autophagy inhibitor that prevents the fusion of autophagosome with lysosome. As expected, autophagy was induced 24 h after hypoxia as demonstrated by a significant increase in staining for LC3 (green) in the presence of chloroquine (Fig. 5b). Hypoxia-induced p62 degradation was blocked in the presence of chloroquine, as well as the reduction of phosphorylated p65/RelA (Fig. 5c and Supplementary Fig. 4A, B). These results point to the fact that hypoxia-induced autophagy is crucial for the down-regulation of proinflammatory signaling. Conversely, hypoxia still reduced the protein expression of NLRP3 in the presence of chloroquine (Fig. 5c and Supplementary Fig. 4C), indicating that NLRP3 degradation is not autophagy-mediated.

**Hypoxia reduces the binding of NLRP3 and mTOR.** Following our results showing that blockage of autophagy in *Il-10*^−/− mice was NLRP3-dependent, and that hypoxia reduces NLRP3

expression concomitantly with the activation of autophagy in the DSS and *Il-10*^−/− mouse models of colitis, as well as cultured IECs, we sought to determine the role of NLRP3 in the regulation of autophagy. HT-29 cells transfected with NLRP3-specific siRNAs displayed a reduction in mTOR phosphorylation and an increased basal autophagy as evidenced by the degradation of the autophagy proteins p62 and LC3 (Fig. 6a and Supplementary Fig. 5A–D), suggesting that NLRP3 plays a crucial role in the constitutive blockage of autophagy through the promotion of mTOR phosphorylation. Moreover, co-immunoprecipitation (Co-IP) experiments identified NLRP3 as a direct binding partner of mTOR and showed that this direct interaction is reversed under hypoxia in IECs (Fig. 6b and Supplementary Fig. 5E, F).

**DMOG reduces DSS colitis and NLRP3, and promotes autophagy.** To study the role of HIF-1α stabilization in hypoxia-mediated anti-inflammatory effects, WT mice were administered with 2% DSS for 7 days in drinking water and treated concomitantly with the hydroxylase inhibitor DMOG. DMOG activates the HIF pathway by mimicking hypoxia through the inhibition of hydroxylase activity, leading to stabilization and transactivation of HIF-1α[9]. DSS-treated mice receiving DMOG presented a less severe colitis compared to mice receiving vehicle as evidenced by a significant reduction in body weight (Fig. 7a) and colonoscopy results (Fig. 7c and d). DMOG-treated mice also showed a clear trend towards less shortening of the colon (Fig. 7b). Hematoxyline and eosin (H&E)-stained sections of distal colon tissue taken from animals treated with DMOG presented a significant lower histological score compared to phosphate-buffered saline (PBS)-treated mice, and showed intact crypts in large areas without extensive infiltration or thickening of

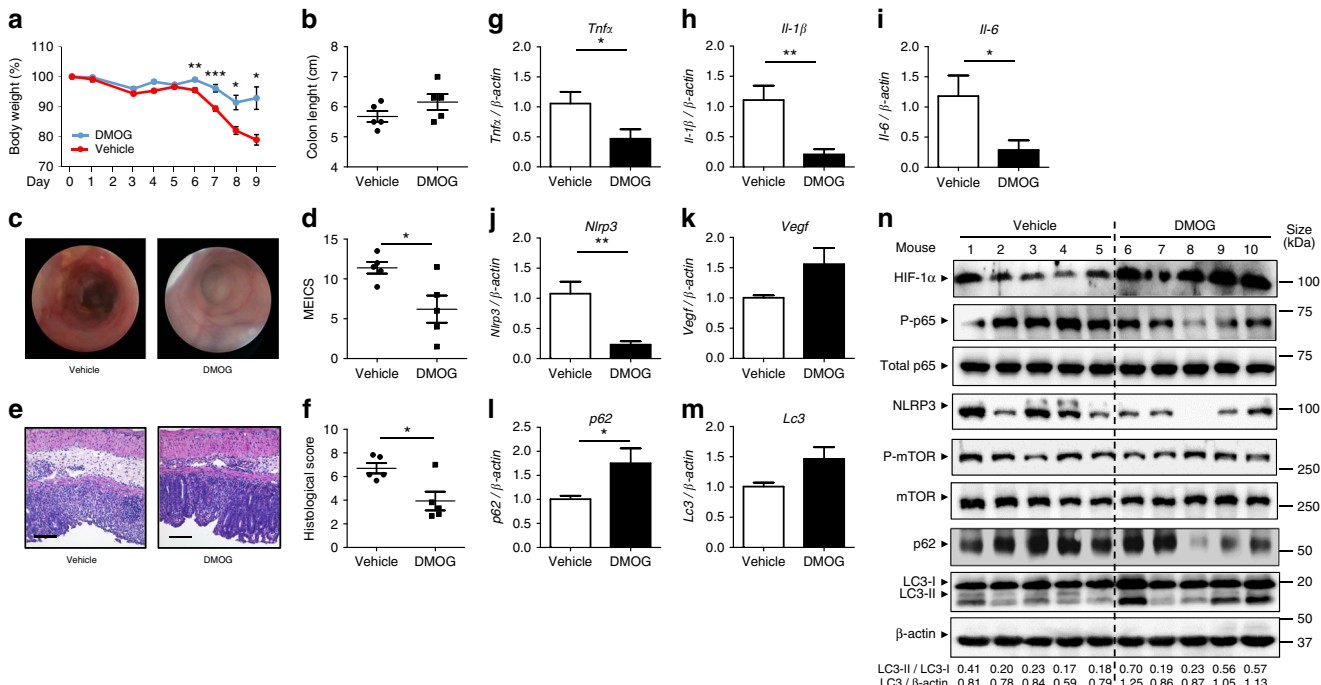

**Fig. 7** DMOG reduces inflammation and induces autophagy in DSS-treated mice. **a**, **b**, **c**, **d**, **e** and **f** WT mice were administered with 2% DSS for 5 days and co-administered with 8 mg DMOG injected intraperitoneally every second day. Weight loss was monitored during the course of treatment **a**. After a recovery period, mice were killed at day 9 colon biopsies were collected and colon length was measured **b**. Mucosal damage was assessed by colonoscopy **c** and murine endoscopic score of colitis severity (*MEICS*) was calculated **d**. H&E staining of distal colon sections displayed a significant decrease in barrier breakdown and inflammatory infiltrate in DMOG-treated mice **e**. Scale bar, 100 μm. Total histological score at day 9 was calculated as the sum of epithelial damage and infiltration score **f**. **g**, **h**, **i**, **j**, **k**, **l** and **m** distal colon biopsies were collected and transcript analysis was performed. Statistical analysis was performed using Student *t*-test.Results represent mean + s.e.m., DSS-treated mice administered with PBS: $n = 5$; DSS-treated mice administered with DMOG: $n = 5$; *$P < 0.05$; **$P < 0.01$. **n** Total protein was isolated and Western blot performed. LC3-I and LC3-II bands were quantified, and autophagy was measured by variations in the ratio of LC3-II/LC3-I and the total amount of LC3 (LC3-I plus LC3-II) relative to β-actin

the mucosa (Fig. 7d and e). DMOG reduced DSS-induced *Tnfa*, *Il-6*, *Il-1β*, and *Nlrp3* mRNA expression (Figs. 7g–j), and induced the expression of the autophagy protein *p62* (Fig. 7l). These mice also presented a tendency towards an increase of *Lc3* (Fig. 7m).At protein level, DMOG-treated mice showed reduced NF-κB phosphorylation and NLRP3 expression, as well as enhanced autophagy evidenced by the increase of the ratio LC3-II/LC3-I and degradation of p62, when compared with vehicle-treated mice (Fig. 7n and Supplementary Fig. 6A–E).

## Discussion

Our results show that environmental hypoxia reduces *TNFα*, *IL-6*, and *NLRP3* expression concomitantly with the induction of *p62* mRNA expression and p62 protein clearance in the colon of patients with CD, suggesting an increase in autophagy turnover[31]. In contrast, biopsies from patients with UC did not display changes in the expression of inflammatory mediators or p62. It is increasingly recognized that CD and UC constitute distinct molecular entities[32]. The discrepancy between patients with CD and UC might reflect the different molecular mechanisms underlying both pathologies. Interestingly, recent studies have shown different HIF expression patterns in intestinal biopsies from patients with UC and CD, suggesting divergent responses to hypoxia[33]. The heterogeneity in disease and treatments might also contribute to the contrasting molecular responses. In line with the results obtained in patients with CD, prolonged hypoxia downregulated NF-κB signaling and the expression of *Tnfa*, *Il-6*, and *Il-1β* in the DSS and *Il-10*$^{-/-}$mouse models of colitis. In HT-29 cells, hypoxia triggered transient accumulation of HIF-1α and resulted in the reduction of basal and LPS-induced NF-κB and

IκBα phosphorylation. These results are in agreement with recent reports showing that hypoxia inhibits LPS-induced *TNFα* and NF-κB activation in colon epithelial cancer cells[34], and confirm previous studies showing protective effects of hypoxia on mucosal inflammation[5, 12, 35]. Conversely, several studies suggest that reduced oxygen levels induce the expression of *TNFα* and *IL-6* in monocytes and directly activate the NF-κB pathway by blocking hydroxylation of factors in that pathway[36, 37].This discrepancy, however, might reflect that responses to hypoxia differ in different cell types. Recent studies indicate that aircraft travel-associated low oxygen partial pressure heightens the risk of flares in IBD patients[4, 38], and altitude-associated hypoxia increases the expression of inflammatory cytokines in the duodenum of healthy individuals[39]. Nonetheless, these effects can be due to confounding factors such as stress, acute exercise, or local changes in circulation[40, 41]. On the other hand, there is mounting evidence that hypoxia promotes resolution of inflammation in several models of disease[5, 13, 42]. In transgenic mice, loss of HIF-1α expression in the epithelium resulted in a more severe colitis, while constitutively active HIF-1 was protective[11, 43, 44].

In line with recent studies linking deficient autophagy with IBD[22, 45, 46], DSS-treated WT mice and *Il-10*$^{-/-}$ mice presented accumulation of phosphorylated/activated mTOR, the main inhibitor of autophagy, and the autophagy protein p62, together with increased NF-κB phosphorylation and inflammatory gene expression. Hypoxia restored autophagy clearance of p62 reducing mTOR phosphorylation and proinflammatory gene expressionand signaling inthese mice. Hypoxia also upregulated *p62* mRNA expression, pointing to a compensatory mechanism for autophagosomal p62 protein degradation. In HT-29 cells,

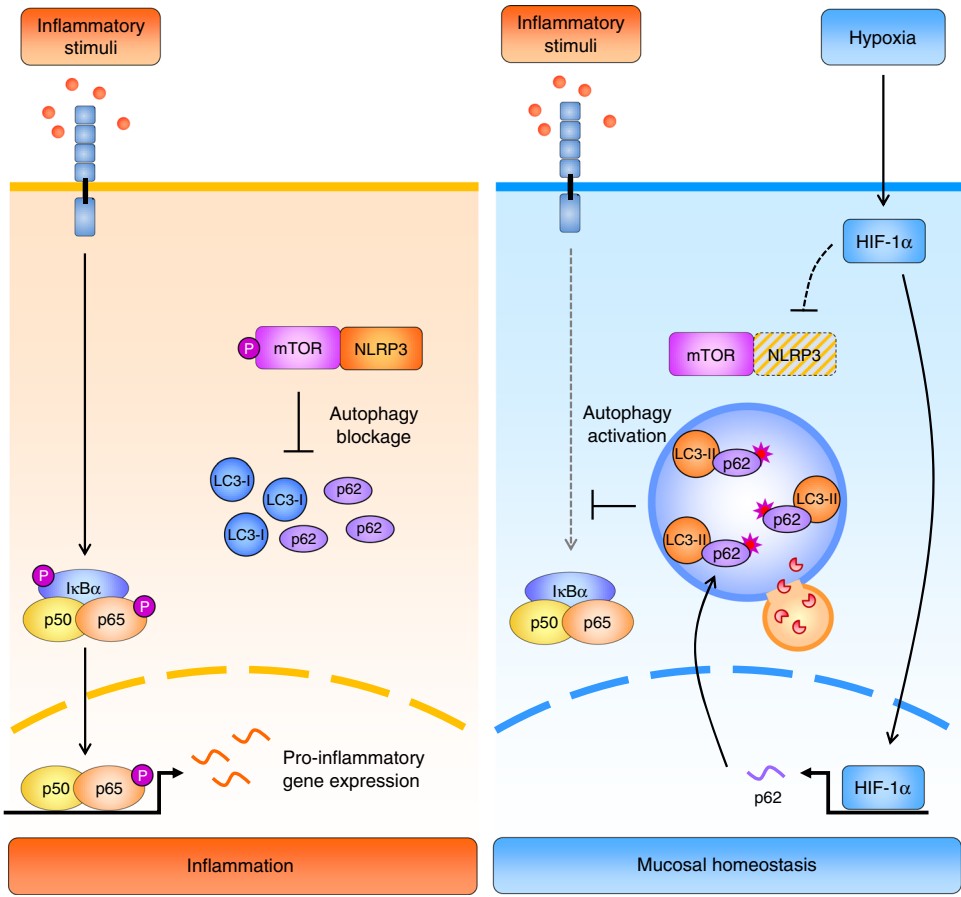

**Fig. 8** Hypoxia ameliorates inflammation through the activation of autophagy via downregulation of NLRP3/mTOR. Under conditions of inflammation, NLRP3 directly binds and promotes the phosphorylation/activation of the autophagy inhibitor mTOR, thereby blocking autophagy. As a result, autophagosomal clearance of pro-inflammatory mediators is impaired, leading to the perpetuation of inflammation. Hypoxia-mediated HIF-1α accumulation induces the expression of p62, and the reduction of NLRP3 protein levels, thereby downregulating mTOR phosphorylation. Subsequent autophagy activation leads to autophagosomal degradation of pro-inflammatory mediators upstream of the NF-κB signalling cascade, leading to reduced proinflammatory gene expression

hypoxia ameliorated inflammatory processes together with the inactivation of mTOR, and induced degradation of p62 and LC3, confirming that hypoxia triggers autophagy through the regulation the catalytic activity of mTOR. This is in good agreement with studies showing that mTOR represses autophagy by phosphorylating, and thereby downregulating the autophagy-associated proteins unc-51-like kinase 1 and ATG13[47, 48]. The blockage of autophagy at late stage with chloroquine reverted hypoxia-mediated inhibition of NF-κB phosphorylation, confirming a causal link between hypoxia-mediated autophagy and the downregulation of inflammation. These results point to an effect upstream of the NF-κB signaling cascade. In this regard, several studies show that autophagy inhibits NF-κB activation through the suppression of the upstream regulatory factor IκB kinase (IKK)[49], and the recruitment of phosphorylated IKK to the autophagosome compartment[50]. Hypoxia-mediated inhibition of NF-κB nuclear translocation is also linked to reduced IKKβ phosphorylation in colon epithelial cells[34]. The two main nuclear factors involved in the regulation of mucosal responses to oxygen deprivation, NF-κB, and HIF-1α, appear to be intimately linked at several mechanistic levels[6]. In line with the significant increase of *p62* mRNA expression observed under hypoxia, ChIP analysis in HT-29 cells showed an increased recruitment of HIF-1α, but not NF-κB, to the promoter of *p62*, indicating a role for HIF-1α in the expression of autophagy proteins. Conversely, NF-κB, but not HIF-1α, was bound to the *NLRP3* promoter under normoxic and

hypoxic conditions. The opposing binding of these nuclear factors to *p62* and *NLRP3* promoters suggest a converse transcriptional regulation of both genes, and appears to align with their antagonistic function. After 24 h of hypoxia, we found lysosomal accumulation in HT-29 cells, but could not detect any effects on active mitochondria. Since lysosomal formation precedes mitochondrial degradation, longer hypoxia periods might be needed to elicit mitochondria damage and clearance.

*Il-10⁻/⁻ Nlrp3⁻/⁻* mice did not present blockage of autophagy under normoxia when compared to *Il-10⁻/⁻* mice, suggesting a role for NLRP3 in the regulation of autophagy. Interestingly, hypoxia downregulated NLRP3 concomitantly with the induction of autophagy in the DSS and *Il-10⁻/⁻* mouse models of colitis, suggesting a role for NLRP3 in hypoxia-induced autophagy. These results were confirmed in HT-29 cells, where hypoxia triggered NLRP3 clearance in a proteasome-independent manner. In contrast to reports showing that autophagy regulates NLRP3 inflammasome function through the autophagosomal degradation of NLRP3[30], hypoxia-mediated NLRP3 clearance appears to be autophagy-independent, pointing to differential regulatory mechanisms for NLRP3 inflammasome-dependent and -independent effects. siRNA-mediated silencing of NLRP3 in HT-29 cells further promoted autophagy clearance of p62 and LC3, confirming a role for NLRP3 in the inhibition of autophagy and reinforcing the idea that NLRP3 degradation precedes and triggers autophagy activation. NLRP3 silencing also reduced the

**Table 1 Participant characteristics**

| Item | HV | CD | UC |
|---|---|---|---|
| Number of patients | 10 | 11 | 9 |
| Gender, females | 3 (30%) | 6 (54.5%) | 5 (55.6%) |
| Age (mean ± SD) | 28.2 ± 4.9 | 35.2 ± 13.3 | 31.8 ± 10.8 |
| Age in males (mean ± SD) | 29 ± 5.5 | 41.3 ± 16 | 30 ± 4.7 |
| Age in females (mean ± SD) | 26.3 ± 3.2 | 29 ± 6.4 | 33.2 ± 14.5 |
| Smoking status | 2/10 (20.0%) | 3/11 (27.3%) | 2/9 (22.2%) |
| *Disease severity* | | | |
| Harvey-Bradshaw Index for CD patients (median, IQR) | NA | 0.0 | NA |
| Partielle Mayo Score for UC patients (median, IQR) | NA | NA | 0.0 |
| *Medical history* | | | |
| Azathioprine / 6-MP | NA | 5/11 (45.5%) | 7/9 (58.3%) |
| Methotrexate | NA | 1/11 (9.1%) | 0 |
| Anti-TNF | NA | 6/11 (54.5%) | 2/9 (16.7%) |
| Systemic steroids | NA | 1/11 (9.1%) | 3/9 (25.0%) |
| NSAID intake | NA | 2/11 (18.2%) | 6/9 (75.0%) |

*CD* Crohn's disease, *HV* Healthy volunteers, *UC* ulcerative colitis, *SD* standard deviation, *NA* not applicable

levels of phosphorylated mTOR, indicating that NLRP3 regulates autophagy upstream of the mTOR signaling pathway. Furthermore, Co-IP experiments identified NLRP3 as a novel binding partner of mTOR. Our results point to that hypoxia-mediated NLRP3 clearance reduces the physical interaction between NLRP3 and mTOR, suggesting that the interaction of both factors is required for mTOR function and subsequent repression of autophagy under normoxia. Inflammasome-independent effects for NLRP3 have been increasingly recognized, and recent research has revealed functions for NLRP3 that do not require the assembly of the inflammasome complex, including the upstream regulation of transforming growth factorβ/Smad signaling[51, 52]. It is well stablished that the catalytic activity of mTOR is tightly regulated by its physical interaction with mTOR complex-binding partners. For example, Raptor regulates mTOR activity by orchestrating the assembly of the mTOR complex and recruiting substrates for mTOR[53, 54]. Hypoxia has been proposed to inhibit mTOR through the regulation of proteins that interfere with its binding with RHEB. Thus, PML protein binds mTOR during hypoxia and inactivate it through the secuestration in nuclear bodies[23]. Likewise, the hypoxia-inducible factor BNIP3 inhibits mTOR activation by interacting with RHEB[55]. Our results suggest that hypoxia-mediated NLRP3 degradation not only counteracts mTOR activation, but might also limit NLRP3 inflammasome-dependent responses trigered by danger signals such as ATP[24]. Notably, hypoxia has been recently discovered to contribute to the control of inflammation, promoting the phosphohydrolysis of extracellular ATP, thereby preventing purinergic signaling[56].

The finding that hypoxia might promote the resolution of inflammation has sparked the interest in the therapeutic use of hydroxylase inhibitors to stabilize HIF for the treatment of IBD. Thus, the hydroxilase inhibitors AKB-4924 and DMOG are protective in the DSS and trinitrobenzene sulfonid acid mouse models of colitis[12, 43]. Our results indicate that DMOG ameliorates DSS colitis and significantly reduces inflammatory gene expression and signaling. Interestingly, HIF-1α stabilization triggered clearance of NLRP3 and p62, and the conversion of LC3-I to LC3-II. These results confirm our findings in the DSS and *Il-10*[−/−] mouse models of colitis, and highlight the important contribution of HIF-1α to hypoxia-mediated autophagy. Recent research has underscored the importance of HIF signaling in controlling inflammatory and innate host defense responses[57]. For example, bacteria- and LPS-mediated induction of HIF-1α has been demonstrated through an oxygen-independent mechanism in myloid and colonic epithelial cells[58–60].

Consequently, manipulating HIF levels has been considered in the context of cancer therapy[61], infectious diseases, and chronic inflammatory conditions such as rheumatoid arthritis and IBD[62].

In summary, our results identify NLRP3 as a binding partner of mTOR and support a novel role for NLRP3 in the regulation of hypoxia-mediated anti-inflammatory effects through NLRP3 clearance, and subsequent autophagy activation and reduction of inflammatory processes in IECs (Fig. 8). Hypoxia-mediated NLRP3 degradation not only prevents stimulatory effects on mTOR signaling but might also limit NLRP3 inflammasome-dependent responses.

## Methods

**Human subjects.** Healthy subjects ($n = 10$), patients with CD ($n = 11$) and patients with UC ($n = 9$) in stable remission, were subjected to hypoxic conditions resembling an altitude of 4000 m.a.s.l. for 3 h using a hypobaric pressure chamber at the Swiss aeromedical center in Dubendorf, Switzerland (ClinicalTrials.gov number, NCT02849821)[63]. Clinical activity in patients with CD and patients with UC was assessed using the Harvey Bradshaw Index and the Partial Mayo Score, respectively. The participant characteristics are shown in Table 1. Distal colon biopsies were taken the day before entering the hypobaric chamber, immediately after hypoxia, and 1 week after collection of the first biopsy at the Division of Gastroenterology and Hepatology of the University Hospital Zurich. This study was approved by the Ethics Committee of the Canton of Zurich (KEK-ZH Nr. 2013-0284) and all participants signed an informed consent.

**Immunohistochemistry.** Intestinal specimens from the distal third of the colon were embedded in paraffin and cut in 5 μm sections with a microtome. Immunostaining for p62 was performed on Leica Bond Max instruments using Refine HRP-Kits (Leica Biosystems, Newcastle, UK). Paraffin-embedded tissue was pretreated for 20 min at 100 °C with citrate-based buffer at pH 6.0 (Leica Biosystems) prior to the incubation with p62 monoclonal antibody (Cat. No. GP62-C; PRO-GEN Biotechnik, Heidelberg, Germany) at a 1:750 dilution. Rabbit anti-Guinea pig (Cat. No. P0141; Dako, Glostrup, Denmark) was used as a secondary antibody at a 1:100 dilution. Immunohistochemistry score was determined by two independent, blinded investigators who assigned score values according to the prevalence of p62 staining (<10% positive cells = 0, 10–25% = 1, 25–50% = 2, 50–75% = 3, and 75–100% = 4)[64].

**Mouse experiments.** Female WT (C57BL/6J) and *Nlrp3*[−/−] mice between 8–10 weeks of age were exposed to 2% DSS[65] (MP Biomedicals) in drinking water *ad libitum* for 7 days. Intestinal inflammation was induced in WT, *Il-10*[−/−], *Nlrp3*[−/−], and *Il-10*[−/−]*Nlrp3*[−/−] mice by oral administration of 1.5% DSS in drinking water *ad libitum* for 5 days[66, 67]. Mice were separated into hypoxia and normoxia groups. The hypoxia group was exposed to hypoxia (8% $O_2$) in a normobaric hypoxic chamber (Coy Laboratory Products, Grass Lake, MI), whereas the normoxia group (21% $O_2$) was used as control. After 18 h of hypoxia, mice were killed for colonic biopsy collection. WT mice were exposed to 2% DSS in drinking water *ad libitum* for 5 days and were co-administered with 8 mg DMOG (Frontier Scientific, Logan, UT) dissolved in PBS, injected intraperitoneally every second day[12]. Control mice received PBS. After a recovery period, mice were killed at day 9 and

distal colon biopsies were taken. Ethical approval for all animal experiments was obtained by the University of Zurich from the Veterinary Office of the Canton of Zurich, Switzerland.

**Colonoscopy and determination of histological score**. Animals were anesthetized intraperitoneally with a mixture of ketamine 90–120 mg/kg body weight (Vétoquinol) and xylazine 8 mg/kg body weight (Bayer), and colonoscopy was performed using the Coloview miniendoscopic system (Karl Storz, Tuttlingen, Germany)[68]. Animals were killed and 1 cm of tissue was removed from the distal third of the colon and fixed in 4% formalin overnight. H&E-stained sections of the paraffin-embedded tissue were used for histological analysis. Histological score was calculated as the sum of epithelial damage and infiltration score[69], and was determined by two independent, blinded investigators.

**RNA isolation and real-time quantitative PCR (RT-qPCR)**. Colonic biopsies were disrupted using the GentleMACS Dissociator (Miltenyi Biotech, Gladbach, Germany). Total RNA from human and mice biopsies, and cells was extracted using RNeasy Plus Kit (Qiagen, Hombrechtikon, Switzerland) according to manufacturer's instructions. For reverse transcription, the High-Capacity cDNA Reverse Transcription Kit (Applied Biosystems, Foster City, CA) was used following the manufacturer's instructions. RT-qPCR was performed using the TaqMan system 7900HT (Applied Biosystems), under the following cycling conditions: 20 s at 95 °C, followed by 40 cycles of 95 °C for 3 s, and 60 °C for 30 s with the TaqMan Fast Universal Mastermix. TaqMan Gene Expression Assays (all from Applied Biosystems) used in this study were human TNF (Hs00174128_m1), IL1B (Hs01555410_m1), IL6 (Hs00174131_m1), NLRP3 (Hs00918082_m1), p62 (SQSTM1) (Hs01061917_g1), LC3 (MAP1LC3A) (Hs01076567_g1) and β-actin Vic TAMRA (4310881E), and mouse Tnf (Mm99999068_m1), Il1b (Mm00434228_m1), Il6 (Mm00446190_m1), Nlrp3 (Mm00840904_m1), Vegfa (Mm00437306_m1), p62 (Sqstm1) (Mm00448091_m1), Lc3 (Map1lc3a) (Mm00458724_m1) and β-actin VIC TAMRA (4352341E). Relative mRNA expression was calculated by the comparative ΔΔCt method using *β-actin* as housekeeping gene.

**Western blotting and Co-IP**. Total protein from biopsies and cells was collected in M-PER lysis buffer (Thermo Fisher Scientific, Reinach, Switzerland). Phospho-p65/RelA (Ser536, 93H1) (Cat. No. 3033), p65/RelA (D14E12) (Cat. No. 8242), phospho-mTOR (Ser2448) (Cat. No. 2971), mTOR (7C10) (Cat. No. 2983), p62 (Cat. No. 5114), phospho-IκBα (Ser32, 14D4) (Cat. No. 2859), IκBα (Cat. No. 9242), ubiquitin (Cat. No. 3933) (Cell Signaling Technology, ZA Leiden, The Netherlands), anti-mouse NLRP3 (Cat. No. AG-20B-0014-C100; AdipoGen, Liestal, Switzerland), anti-mouse HIF-1α (EP1215Y) (Cat. No. ab51608; Abcam, Cambridge, UK), anti-human HIF-1α (Cat. No. NB100-479; Novus Biologicals, Littletown, CO), anti human NLRP3 (Cat. No. ALX-804-819-C100; Enzo Life Sciences, Lausen, Switzerland), LC3 (Cat. No. L8919) and β-actin (Cat. No. A5441) (Sigma-Aldrich, St Louis, MO) antibodies were used at 1:1000 dilution for western blot. Co-IP was performed overnight at 4 °C using the human NLRP3 antibody at 1:100 dilution. Immunocomplexes were collected with protein G sepharose beads (GE Healthcare, Glattburgg, Switzerland) for 1 h at 4 °C prior to Western blotting. Densitometry of bands was measured using ImageJ software.

**Cell culture and hypoxia treatment**. HT-29 cells were obtained from the German Collection of Cells and Microorganisms (DSMZ, Braunschweig, Germany), and cultured in DMEM medium supplemented with 10% FCS (VWR, Dietikon, Switzerland). In some experiments, HT-29 cells were treated with 10 μg/ml LPS (Invitrogen, Carlsbad, CA). Late-stage autophagy was inhibited with 40 μM chloroquine (Sigma-Aldrich) for 1 h prior to hypoxia. Proteasomal activity was inhibited with 20 μM MG132 (Sigma-Aldrich) for 1 h prior to 24 h hypoxia. Cells were exposed to hypoxia (0.2% $O_2$) in a hypoxia workstation incubator (In vivo 400, Baker Ruskinn, Bridgend, UK). Cells maintained in normoxia (21% $O_2$) for the same time-period were used as controls.

**Confocal microscopy**. After 24 h under normoxia or hypoxia, HT-29 cells were stained with LysoTracker Yellow-HCK-123 and MitoTracker Deep Red FM (Thermo Fisher Scientific) according to manufacturer's instructions. For LC3 staining, HT-29 cells subjected to normoxia or hypoxia for 24 h were fixed with 4% paraformaldehyde for 20 min and then permeabilized in 100% methanol. After blocking with 3% bovine serum albumin, cells were incubated with LC3 antibody (Cat. No. PM036; MBLI, Woburn, MA) at 1:200 dilution overnight at 4 °C. Cells were then incubated with Alexa 488 conjugated secondary rabbit antibody (Invitrogen) for 1 h and DAPI for 5 min before mounting with antifade medium (Dako). Cells were analysed by a Leica SP5 laser scanning confocal microscope (Leica Microsystems, Wetzlar, Germany). Fluorescence images of the same sample were acquired and processed using Leica confocal software (LAS-AF Lite; Leica Microsystems). Quantification of LC3/DAPI was performed by two independent, blinded investigators using ImageJ software in at least ten areas of interest pooled from four independent experiments.

**Small interfering (siRNA) transfection**. HT-29 cells were transfected with 10 pmol of three sets of siRNAs for NLRP3 (Life Technologies) using Lipofectamine RNAiMAX (Invitrogen) according to the manufacturer's instructions.

**ChIP analysis**. ChIP was performed using the ChIP-IT Express Enzymatic kit (Active Motif, Carlsbad, CA) according to the manufacturer's instructions. Immunoprecipitation of 25 μg of DNA was carried out overnight at 4 °C using anti-HIF-1α (Cat. No. NB100-479; Novus biologicals, Littletown, CO) or anti-p65/RelA (D14E12) (Cat. No. 8242; Cell Signaling Technology) antibodies at a concentration of 1 μg/ml. DNA isolated from an aliquot of the total nuclear extract was used as a loading control for the PCR (input control). PCR was performed with total DNA and immunoprecipitated DNA using the following promoter-specific primers, p62 promoter-binding site for HIF-1α: 5′-ATCCCGATCGGCTCCCG-3′ (forward), 5′-ACTCCCACTCCAGCCTTACC-3′ (reverse) (70 bp), and NF-κB: 5′-ACTGGGCGGTTGAAGCC-3′ (forward), 5′-TCCAGAGGCCCAGAGGAAG-3′ (reverse) (87 bp), NLRP3 promoter-binding site for HIF-1α: 5′-TTCTTCTGGCGTCTCCAAGG-3′ (forward), 5′-TTGCACACGTA-GAGCCCAAA-3′ (reverse) (81 bp), and NF-κB: 5′-CCAGTATGGAACCGAGA-CACG-3′ (forward), 5′-CAGAGCAGAGGCAGAGAGACT-3′ (reverse) (97 bp). The PCR products (10 μl) were subjected to electrophoresis on a 2% agarose gel.

**Statistics**. Statistical analysis was performed using Student *t* test, or one-way ANOVA followed by Tukey's post-test. The results are expressed as mean + standard error of the mean (s.e.m.) and significance was set as $p < 0.05$.

**Data availability**. The authors declare that all the data supporting the findings of this study are available in the manuscript or are available from the corresponding author upon reasonable request.

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

## Acknowledgements

We thank Mirjam Blattmann and Sylvie Scharl for their organizing and collecting the human colon biopsies. Mehdi Madanchi is gratefully acknowledged for his help in collecting the human subject data. We extend our special appreciation to Silvia Lang for her technical support, and the Zurich Integrative Rodent Physiology (ZIRP) core facility of the University of Zurich for their support in the animal experiments. This research was supported by a grant from the Swiss Philanthropy Foundation to GR and research grants from the Swiss National Science Foundation to GR (Grant No. 324730_138291 and 314730_153380), and the Swiss IBD Cohort (Grant No. 314730_153380). J.C.-R. was supported by the European Crohn's and Colitis Organisation (ECCO) Fellowship.

## Author contributions

J.C.-R., S.S., H.M., K.A., M.H., M.R.S., P.S., S.R.V., and J.Z.: Data acquisition; I.F.-W., C.dV, and R.H.W.: Critical revision; GR: Study supervision, manuscript preparation: P.A.R.: Study supervision, data acquisition, manuscript preparation.

## Additional information

**Competing interests:** The authors declare no competing financial interests.

