## [Peer Review file · Nature Communications]

Reviewers' comments:

Reviewer #1 (Remarks to the Author):

In this manuscript Cosin-Roger et al describe a mechanism by which hypoxia activates autophagy through the targeting of NLRP3 through a non-autophagic degradation mechanism. It has been reported by others that hypoxia is sufficient to induce autophagy and subsequent autophagic degradation of p62. The role of hypoxia in the intestine has been controversial. Much of the novelty of this manuscript relies on the link between NLRP3/autophagy/mTOR and the link between this pathway and intestinal inflammation. This reviewer believes that the paper remains premature. The most significant concerns are 1) the significance of the connection between WT HT29 cells and the IL10^{-/-} mice 2) the lack of data on intestinal inflammation even though intestinal inflammation is included in the title of the manuscript.

Major Issues:

Figure 1: The authors include data from IBD patients which is a heroic effort. The changes in inflammatory cytokines in these samples are very clear. My biggest concern is the assessment of autophagy by mRNA analysis. Currently it is unclear whether the p62/LC3 data suggests a block or an induction in autophagy. It would be good to try to assess protein for p62 and/or LC3II/I ratios as the authors did in other figures in the manuscript. This is clearly a challenging experiment and might not be technically feasible.

Figure 2: The authors mentioned treating mice with DSS however it was unclear if all panels in figure 2 were post-DSS treatment. One challenge in interpreting this data is that IL10^{-/-} are likely more susceptible to DSS and thus should have increased inflammatory infiltration in the intestine. Without histologic or FACS assessment of the intestine it is impossible to determine the effects of these treatments on intestinal inflammation. Additionally, the high levels of p62 in WT mice and lack of change in these levels under hypoxic conditions suggests that the effect is limited to IL10^{-/-} mice. Therefore it is unclear how this data is connected to the patient data or the data in HT29 cells which uses WT cells.

Figure 3: No time point is given for the MG132 treatment suggesting that NLRP3 may be degraded via the proteasome. Would it be helpful to perform a time course or include a positive control? I'm having trouble connecting the ChIP data to the mechanistic analysis. Much of the data remains correlative.

Figure 4: In figure 4A it is unclear why there are so few lysosomes. Quantification would be helpful for Figure 4A. In figure 4B the authors should list how many images were scored, whether the samples were blinded for analysis. For quantification # of LC3 puncta is a standard assessment. Figure 4C does not seem to support the authors hypothesis. It seems that the increase in p62 with and without chloroquine is relatively similar under normoxic conditions and during hypoxia suggesting similar levels of autophagic turnover.

Figure 5: Figure 5A quantification of LCII/I ratios should be included for multiple independent experiments. The shifts observed in this figure are quite small thus the role of NLRP3 is difficult to conclude.

Reviewer #2 (Remarks to the Author):

This study describes a novel hypoxia-NLRP3 axis in the regulation of intestinal mucosal autophagic responses. Autophagy has recently emerged as a key pathway in inflammatory bowel disease (IBD), and hypoxia signaling has been shown to mediate both protective and detrimental effects in the context of intestinal inflammation a number of studies. Utilizing both IBD patient biopsies and

an IL10 knockout mouse colitis model, Cosin-Rogers et al. demonstrate that exposure to hypoxia abrogates pro-inflammatory cytokine expression and induces autophagy-associated gene expression in an NLRP3-dependent manner. The authors further identify NLRP3 as a binding partner of the autophagy master regulator mammalian target of rapamycin (mTOR), defining a novel mechanism for autophagy induction by hypoxia-mediated NLRP3 degradation. This is a comprehensive and innovative study with important implications for hypoxic regulation of mucosal inflammatory processes. A number of additional issues should be addressed:

Major points:

1. The authors use distal colon biopsies from hypoxia-exposed CD and UC patients, and demonstrate statistically significant changes in cytokine and autophagy gene expression almost exclusively in CD patients (Figure 1). Did the authors perform histological assessment of patient biopsies? Were these inflamed or non-inflamed regions?
2. In a related question, can the authors speculate as to the apparent disease selectivity of these findings?
3. What happens to expression of NLRP3 in the inflamed IL10 deficient mouse colon? Are there changes in NLRP3 levels compared to non-DSS treated mice?
4. Is colonic NLRP3 expression similarly reduced in vivo upon hypoxia exposure?
5. The ChIP-PCR results shown in Figure 3 correspond to 24 hours of hypoxia exposure. Given the acute timeline of hypoxia treatment and observed HIF-1 α stabilization in other studies (see especially Figure 3A), this seems excessive. The authors should consider profiling HIF-1/p65 binding to p62 and NLRP3 promoter regions at an earlier time-point.
6. In Figure 4A, the authors suggest that active mitochondria are unaltered after 24 hour hypoxia treatment. However, the representative images of MitoTracker Deep Red FM staining do not convincingly portray this, and arguably demonstrate reduced staining and/or altered morphology. This should be more closely quantified.
7. Relevant to data outlined in Figure 5 – have the authors monitored mTOR and p-mTOR protein levels in normoxic vs hypoxic HT-29 cells?

Minor Points:

1. Page 14, paragraph 1: reference 53 is missing.

Reviewer #3 (Remarks to the Author):

In the present studies, the authors provide evidence for an anti-inflammatory role for hypoxia signaling during intestinal inflammation via downregulation of the NLRP3 inflammasome. Overall, these are very exciting findings. I think additional in vivo studies with pharmacologic or genetic modulation of the HIF pathway would further strengthen an already outstanding manuscript.

Major comments:

- 1.) The authors may consider to provide in vivo data linking their proposed pathway more directly to HIF, e.g. by utilizing in vivo HIF activators or genetic models of HIF deletion.

Minor comments:

1.) The authors should expand their discussion to consider more chronic disease conditions and the role of HIFs in innate immunity (e.g. PMID: 22988108)

2.) Along the same lines - the authors may consider discussing a link between hypoxia, purinergic signaling and IBD with regard to their findings on NLRP3 repression (e.g. PMID: 25850656 and PMID: 21942704)

Reviewer #1:

Major Issues:

1) Figure 1: The authors include data from IBD patients which is a heroic effort. The changes in inflammatory cytokines in these samples are very clear. My biggest concern is the assessment of autophagy by mRNA analysis. Currently it is unclear whether the p62/LC3 data suggests a block or an induction in autophagy. It would be good to try to assess protein for p62 and/or LC3II/I ratios as the authors did in other figures in the manuscript. This is clearly a challenging experiment and might not be technically feasible.

Response: We agree with the reviewer that autophagy cannot be assessed by mRNA analysis, and autophagic protein degradation is required to establish induction of autophagy. Consequently, we performed immunohistochemistry analysis for p62 in colonic biopsies of healthy volunteers, patients with CD and patients with UC at the designated time points. Although immunohistochemistry score did not show significant differences between samples taken prior to and after hypoxia, there is a trend towards a reduction in the levels of p62 one week after hypoxia in healthy subjects and patients with CD, indicating an increased rate of autophagic turnover. This is in accordance with the results obtained in our experiments with the DSS and IL-10^{-/-} mouse models of colitis and cultured cells. We have expanded Figure 1 showing representative immunohistochemistry pictures, as well as the immunohistochemistry score (Figure 1G), and have amended the Results section accordingly, which now reads:

“immunostaining for p62 of distal colon sections showed a trend towards reduced protein levels of p62 one week after hypoxia in HV and patients with CD, indicating a hypoxia-mediated increase in autophagy flux (Figure 1G).”

2) Figure 2: The authors mentioned treating mice with DSS however it was unclear if all panels in figure 2 were post-DSS treatment. One challenge in interpreting this data is that IL10^{-/-} are likely more susceptible to DSS and thus should have increased inflammatory infiltration in the intestine. Without histologic or FACS assessment of the intestine it is impossible to determine the effects of these treatments on intestinal inflammation. Additionally, the high levels of p62 in WT mice and lack of change in these levels under hypoxic conditions suggests that the effect is limited to IL10^{-/-} mice. Therefore it is unclear how this data is connected to the patient data or the data in HT29 cells which uses WT cells.

Response: We agree that results from IL-10^{-/-} mice pre-treated with DSS are difficult to interpret and further validation in a different IBD animal model was needed. Consequently, we have now repeated the hypoxia experiments using the DSS-mouse model of colitis. Specifically, WT and *Nlrp3*^{-/-} mice were exposed to 2% DSS or DSS-free water for seven days and subjected to normoxic or hypoxic conditions (Figure 2). Additionally, DSS-treated WT mice were administered concomitantly with the hydroxylase inhibitor DMOG to mimic hypoxia (Figure 7). The new data confirm the previous results showing that hypoxia downregulates NLRP3 and inflammatory processes concomitantly with the induction of autophagy.

We also agree that the results obtained in the experiment using the IL-10^{-/-} mouse model suggest that this effect is limited to IL-10^{-/-} mice. While we do not have a clear explanation for this, new results show that hypoxia reduces protein levels of p62 in both H₂O- and DSS- treated WT mice (Figure 2), as well as in most WT mice treated with DMOG (Figure 7). This is in accordance with our data in WT HT-29 cells in the presence or absence of LPS (Figure 4) and the new immunohistochemistry analysis performed in patient samples, which shows a trend towards a decrease of p62 in healthy volunteers and patients

with CD (Figure 1). Taken together, these results point to a hypoxia-mediated reduction of p62 levels under both inflamed and non-inflamed conditions.

3) Figure 3: No time point is given for the MG132 treatment suggesting that NLRP3 may be degraded via the proteasome. Would it be helpful to perform a time course or include a positive control? I'm having trouble connecting the ChIP data to the mechanistic analysis. Much of the data remains correlative.

Response: We would like to thank the reviewer for the note and we have now added the time of treatment with MG132 to the Methods section, which now reads:

“Proteasomal activity was inhibited with 20 μ M MG132 (Sigma-Aldrich) for 1 h prior to 24 h hypoxia.”

We also agree that a positive control to confirm proteasomal inhibition in the presence of MG132 during hypoxia would be an important addition. Accordingly, we have now performed Western blotting for ubiquitin (now Figure 4E). Our results show an accumulation of ubiquitinated proteins in the presence of MG132, thereby confirming proteasomal blockage during the experiment. The newly generated data also show that hypoxia reverses ubiquitinated protein accumulation even in the presence of MG132, suggesting hypoxia-mediated activation of additional protein degrading mechanisms, including autophagy. We have amended the Results section, which now reads:

“hypoxia reversed the accumulation of ubiquitinated proteins in the presence of the proteasome inhibitor, pointing to the activation of alternative protein degradation mechanisms under hypoxia, such as autophagy (Figure 4E).”

We fully agree that ChIP data were not well connected to the rest of the results and the proposed mechanism. We have now repeated ChIP analysis for HIF-1 and p65 at an earlier time point (6 h) corresponding to the time point at which HIF-1 peaks. Results show increased binding of HIF to the p62 promoter after 6h of hypoxia (Figure 3F) and suggest a link between hypoxia-mediated HIF stabilization/transcriptional activation and the induction of p62 mRNA expression. The important contribution of HIF-1 α to the proposed mechanism was also evidenced in DSS-treated WT mice administered with DMOG, where DMOG-mediated HIF accumulation triggered NLRP3 clearance and autophagy (Figure 7).

4) Figure 4: In figure 4A it is unclear why there are so few lysosomes. Quantification would be helpful for Figure 4A. In figure 4B the authors should list how many images were scored, whether the samples were blinded for analysis. For quantification # of LC3 puncta is a standard assessment. Figure 4C does not seem to support the authors hypothesis. It seems that the increase in p62 with and without chloroquine is relatively similar under normoxic conditions and during hypoxia suggesting similar levels of autophagic turnover.

Response: The reviewer certainly is right and quantification of images in Figure 4A (now Figure 5A) would be an important addition. In order to clarify the previous results, we have now repeated the experiment and performed lysosomal and mitochondrial quantification using DAPI staining for normalization. We have also increased the number and magnification of images shown in the manuscript. Quantification of LysoTracker staining relative to DAPI confirms our previous data supporting an accumulation of lysosomes under hypoxia (Figure 5A). We have amended the figure legend, which now reads:

“HT-29 cells were subjected to hypoxia (0.2% O₂, 24 h) and stained with DAPI (blue), as well as LysoTracker Yellow-HCK-123 (yellow) or MitoTracker Deep Red FM

(red) to monitor lysosomal accumulation and functional mitochondria, respectively. Examples of cells containing both markers are presented in the magnified insets. Scale bar, 10 μ m. Changes in lysosomal accumulation and functional mitochondria were calculated relative to DAPI staining from ten areas of interest pooled from 4 independent experiments by two independent, blinded investigators. Results represent mean \pm SEM, **, $p < 0.01$."

We would like to thank the reviewer for addressing the lack of information in Figure 4B (now Figure 5B). We have amended the Figure legend and the Methods section, which now reads:

"Quantification of LC3/DAPI was performed by two independent, blinded investigators using ImageJ software in at least ten areas of interest pooled from four independent experiments."

We quantified LC3 fluorescent dots using ImageJ software, a well established method regularly used in publications in the field of autophagy, and as previously described (Guo et al., *Autophagy*, 2015; Gerster et al., *CMGH*, 2016).

The immunoblot for p62 in Figure 4C (now Figure 5C) was included as a positive control to confirm blockage of autophagy in the presence of chloroquine. Chloroquine is a late-stage autophagy inhibitor that prevents hypoxia-induced p62 degradation. Hence, levels of p62 should be similar in both normoxic and hypoxic conditions in the presence of chloroquine, as reflected in Figure 5C.

5) Figure 5: Figure 5A quantification of LCII/I ratios should be included for multiple independent experiments. The shifts observed in this figure are quite small thus the role of NLRP3 is difficult to conclude.

Response: We agree with the reviewer that quantification from multiple experiments would better clarify the direct implication of NLRP3 in the induction of autophagy in Figure 5A (now Figure 6A). We have performed new siRNA transfections and Western blots, and repeated the quantification of LC3-II/LC3-I and LC3/ β -actin ratios. Our results show a significant decrease in the ratio LC3/ β -actin in the presence of NLRP3 siRNA. Although not significant, our results also show a trend towards an increase in the ratio LC3-II/LC3-I (Figure 6A). Additionally, Western blot analysis showed a clear decrease in p62 and phospho-mTOR in cells transfected with NLRP3 siRNA. Taken together our results suggest a role of NLRP3 in the regulation of autophagy.

Reviewer #2:

Major points:

1. The authors use distal colon biopsies from hypoxia-exposed CD and UC patients, and demonstrate statistically significant changes in cytokine and autophagy gene expression almost exclusively in CD patients (Figure 1). Did the authors perform histological assessment of patient biopsies? Were these inflamed or non-inflamed regions?

Response: We have now performed immunostaining for p62 in histology samples from healthy subjects, patients with CD and patients with UC. Although we detected changes at molecular level in inflammatory expression, histology samples did not show any differences in the inflammatory status prior to or after hypoxia (Figure 1G). This might be explained by the fact that IBD patients were in remission (see table 1) and 3 hours of hypoxia are not likely to induce profound histological changes. Samples were collected from macroscopically

non-inflamed regions, which might also explain the lack of marked histologic features of inflammation in these samples. Taken into consideration the new data, we have amended the Results section, which now reads:

“Although histology samples did not show any differences in the inflammatory status of patients with IBD prior to or after hypoxia, immunostaining for p62 of distal colon sections showed a trend towards reduced protein levels of p62 one week after hypoxia in HV and patients with CD”

2. In a related question, can the authors speculate as to the apparent disease selectivity of these findings?

Response: This is indeed a very interesting point and we agree it needs to be addressed. While the reasons for the differences between samples from CD and UC patients are not clear, divergences in inflammatory gene expression might reflect differences in molecular mechanisms underlying CD and UC. We have addressed this discrepancy in the Discussion section, which now reads:

“In contrast, biopsies from patients with UC did not display changes in the expression of inflammatory mediators or p62. It is increasingly recognized that CD and UC constitute distinct molecular entities.³¹ The discrepancy between patients with CD and UC might reflect the different molecular mechanisms underlying both pathologies. Interestingly, recent studies have shown different HIF expression patterns in intestinal biopsies from patients with UC and CD suggesting divergent responses to hypoxia.³² The heterogeneity in disease and treatments might also contribute to the contrasting molecular responses”.

3. What happens to expression of NLRP3 in the inflamed IL10 deficient mouse colon? Are there changes in NLRP3 levels compared to non-DSS treated mice?

Response: In this updated version of the manuscript, we have included Taqman and Western blot analysis for NLRP3 in all previous and new animal experiments. In IL-10 deficient mice (Figure 3, D and H) and DSS-treated WT mice (Figure 2, D and H), we found increased expression of NLRP3 at mRNA and protein level when compared to non-inflamed mice, which was significantly reduced under hypoxia. Similar results were obtained in DSS-treated WT mice administered with the hydroxylase inhibitor DMOG to mimic hypoxia (Figure 7, J and N).

4. Is colonic NLRP3 expression similarly reduced in vivo upon hypoxia exposure?

Response: As explained above, and consistent with our previous results in IBD patients (Figure 1A) and cultured epithelial cells (figure 4, A and C), the new data show that hypoxia reduces NLRP3 expression in IL-10^{-/-} and DSS-treated WT mice (Figure 3, D and H; and Figure 2, D and H), confirming hypoxia-mediated downregulation of NLRP3.

5. The ChIP-PCR results shown in Figure 3 correspond to 24 hours of hypoxia exposure. Given the acute timeline of hypoxia treatment and observed HIF-1alpha stabilization in other studies (see especially Figure 3A), this seems excessive. The authors should consider profiling HIF-1/p65 binding to p62 and NLRP3 promoter regions at an earlier time-point.

Response: We fully agree with the reviewer that the time point chosen for ChIP analysis was excessive. As suggested by the reviewer, we have profiled HIF-1 α /p65 binding to p62 and NLRP3 promoters at an earlier time point (6 h) corresponding to the time point at which

HIF-1 α accumulation peaks (see updated Figure 4A). Results show increased binding of HIF to the p62 promoter after 6h of hypoxia (Figure 4, F and G). Consequently, we have amended the Results section, which now reads:

“In line with the stabilization of HIF-1 α after 6 h of hypoxia, ChIP analysis shows increased binding of HIF-1 α to the promoter of p62 after 6 h of hypoxia compared to normoxia, suggesting a role for HIF-1 α in the induction of p62 expression observed under low oxygen level conditions (Figure 4F).”

6. *In Figure 4A, the authors suggest that active mitochondria are unaltered after 24 hour hypoxia treatment. However, the representative images of MitoTracker Deep Red FM staining do not convincingly portray this, and arguably demonstrate reduced staining and/or altered morphology. This should be more closely quantified.*

Response: We agree that results depicted in Figure 4A (now Figure 5A) were not overly convincing. As stated above we have now repeated the experiment and performed lysosomal and mitochondrial quantification using DAPI staining for normalization. We have also increased the number and magnification of images. Quantification of Mitotracker staining relative to DAPI confirms our previous data suggesting no observable changes in active mitochondria. The amended figure legend now reads:

“HT-29 cells were subjected to hypoxia (0.2% O₂, 24 h) and stained with DAPI (blue), as well as LysoTracker Yellow-HCK-123 (yellow) or MitoTracker Deep Red FM (red) to monitor lysosomal accumulation and functional mitochondria, respectively. Examples of cells containing both markers are presented in the magnified insets. Scale bar, 10 μ m. Changes in lysosomal accumulation and functional mitochondria were calculated relative to DAPI staining from ten areas of interest pooled from 4 independent experiments by two independent, blinded investigators. Results represent mean \pm SEM, **, p < 0.01.”

7. *Relevant to data outlined in Figure 5 – have the authors monitored mTOR and p-mTOR protein levels in normoxic vs hypoxic HT-29 cells?*

Response: in our previous manuscript, we assessed mTOR and p-mTOR protein levels in normoxic vs hypoxic HT-29 cells in Figure 3A (now in Figure 4A). Our results showed that hypoxia reduces mTOR phosphorylation in HT-29 cells in the presence or absence of LPS. In this updated version of the manuscript, we have included Western blot analysis for p-mTOR and total mTOR in all animal experiments showing a reduction of p-mTOR under hypoxia in the DSS and IL-10 $^{-/-}$ mouse models of colitis (Figures 2H and 3H) and pointing to the induction of autophagy under hypoxic conditions *in vivo* and *in vitro*.

Minor Points:

1. *Page 14, paragraph 1: reference 53 is missing.*

Response: The missing reference has been included.

Reviewer #3:

Major comments:

1.) *The authors may consider to provide in vivo data linking their proposed pathway more*

directly to HIF, e.g. by utilizing in vivo HIF activators or genetic models of HIF deletion.

Response: As suggested by the reviewer, we have now performed an additional *in vivo* experiment using DSS-treated WT mice administered with the HIF activator dimethyloxallylglycine (DMOG) (Figure 7). Our results show that DMOG ameliorates DSS-induced colitis concomitantly with the clearance of NLRP3 and activation of autophagy, thereby confirming the protective effects of hypoxia observed in our previous and new *in vivo* and *in vitro* experiments. Importantly, the new results have significantly contributed to shed light on the proposed mechanism highlighting the important contribution of HIF-1 α to hypoxia-mediated NLRP3 degradation and the induction of autophagy.

Minor comments:

1.) The authors should expand their discussion to consider more chronic disease conditions and the role of HIFs in innate immunity (e.g. PMID: 22988108)

Response: Given the new findings on the pivotal role of HIF-1 α in the proposed mechanisms, we have expanded the Discussion on the role of HIF in innate immune responses and the therapeutic potential of HIF regulation in chronic inflammatory conditions as suggested, which now reads:

“Recent research has underscored the importance of HIF signaling in controlling inflammatory and innate host defense responses.⁵⁵ For example, bacteria- and LPS-mediated induction of HIF-1 α has been demonstrated through an oxygen-independent mechanism in myeloid and colonic epithelial cells.^{56, 57, 58} As a result, manipulating HIF levels has been considered in the context of cancer therapy⁵⁹, infectious diseases and chronic inflammatory conditions, such as rheumatoid arthritis and IBD.⁶⁰”

2.) Along the same lines - the authors may consider discussing a link between hypoxia, purinergic signaling and IBD with regard to their findings on NLRP3 repression (e.g. PMID: 25850656 and PMID: 21942704)

Response: As suggested by the reviewer, we have commented the impact of the presented mechanisms on purinergic signaling in the discussion, which now reads:

“Our results suggest that hypoxia-mediated NLRP3 degradation not only counteracts mTOR activation, but might also limit NLRP3 inflammasome-dependent responses triggered by danger signals such as ATP.²⁴ Notably, hypoxia has been recently discovered to contribute to the control of inflammation promoting the phosphohydrolysis of extracellular ATP, thereby preventing purinergic signaling.⁵⁴”

REVIEWERS' COMMENTS:

Reviewer #1 (Remarks to the Author):

The authors have done a nice job of addressing all the reviewer comments. I believe the manuscript is suitable for publication.

Reviewer #2 (Remarks to the Author):

The authors have made a commendable effort to address all concerns, and the manuscript is greatly improved as a result. No additional concerns are noted.

Reviewer #3 (Remarks to the Author):

Thank you for considering my comments.

REVIEWERS' COMMENTS:

Reviewer #1 (Remarks to the Author):

The authors have done a nice job of addressing all the reviewer comments. I believe the manuscript is suitable for publication.

Reviewer #2 (Remarks to the Author):

The authors have made a commendable effort to address all concerns, and the manuscript is greatly improved as a result. No additional concerns are noted.

Reviewer #3 (Remarks to the Author):

Thank you for considering my comments.

There are no further issues raised by the reviewers. We would like to thank the reviewers for their comments and suggestions for the manuscript during the revision period.